# Vertical Jump Data from Inertial and Optical Motion Tracking Systems

Mateo Rico-Garcia [1], Juan Botero-Valencia [2] and Ruber Hernández-García [3,*]

1. Facultad de Ingeniería, Institución Universitaria Pascual Bravo, Calle 73 No. 73A-226, Medellin 050034, Colombia
2. Grupo Sistemas de Control y Robótica, Facultad de Ingenierías, Instituto Tecnológico Metropolitano—ITM, Calle 73 No. 76A-354, Medellin 050034, Colombia
3. Research Center for Advanced Studies of Maule (CIEAM), Universidad Católica del Maule, Avenida San Miguel 3605, Talca 3480094, Chile
* Correspondence: rhernandez@ucm.cl

**Abstract:** Motion capture (MOCAP) is a widely used technique to record human, animal, and object movement for various applications such as animation, biomechanical assessment, and control systems. Different systems have been proposed based on diverse technologies, such as visible light cameras, infrared cameras with passive or active markers, inertial systems, or goniometer-based systems. Each system has pros and cons that make it usable in different scenarios. This paper presents a dataset that combines Optical Motion and Inertial Systems, capturing a well-known sports movement as the vertical jump. As a reference system, the optical motion capture consists of six Flex 3 Optitrack cameras with 100 FPS. On the other hand, we developed an inertial system consisting of seven custom-made devices based on the IMU MPU-9250, which includes a three-axis magnetometer, accelerometer and gyroscope, and an embedded Digital Motion Processor (DMP) attached to a microcontroller mounted on a Teensy 3.2 with an ARM Cortex-M4 processor with wireless operation using Bluetooth. The purpose of taking IMU data with a low-cost and customized system is the deployment of applications that can be performed with similar hardware and can be adjusted to different areas. The developed measurement system is flexible, and the acquisition format and enclosure can be customized. The proposed dataset comprises eight jumps recorded from four healthy humans using both systems. Experimental results on the dataset show two usage examples for measuring joint angles and COM position. The proposed dataset is publicly available online and can be used in comparative algorithms, biomechanical studies, skeleton reconstruction, sensor fusion techniques, or machine learning models.

**Keywords:** biomechanics; inertial measurement units-imu; motion capture system-mocap; sports; vertical jump

## 1. Introduction

Motion capture (MOCAP) is a technique that started around the 1970s using photogrammetry for biomechanical analysis. It is used in various fields such as education [1], sports training [2], and cinema and video games [3]. Its main purpose is to record the movements of an object, in this case the human body, for later analysis and to apply data modeling [4,5]. In human motion capture, it is required to establish the position and orientation of each of the body segments [6–8]. Technological development has resulted in various systems for measuring the position of body segments and angles, most notably optical and Inertial Measurement Units (IMU). Optical capture systems use a set

of cameras and reflective markers to obtain the position from the superposition of the images. Its greatest advantage is the accuracy that, in the case of the system used in this work, reaches ±0.1 mm. IMU-based capture systems use a combination of accelerometers, gyroscopes, and magnetometers. Inertial systems have an advantage over optical systems because they are portable and can be used in environments with uncontrolled conditions. Moreover, in comparison, implementations such as the one presented in this paper can be lower-cost options.

One of the main uses of IMUs is related to orientation determination. Different works have been developed supported by datasets, such as in [9], where inertial orientation estimation (IOE) is validated by providing a large reference dataset. In addition, ref. [10] presented examples of activities of daily living (ADL), and [11] developed a deep-learning-based method for gait recognition from IMUs' raw information. A current trend in the biomechanical analysis is to mix or fuse data to derive measurements or find patterns, such as [12] where IMUs, pressure-distribution, and photoelectric data are obtained for gait analysis.

On the other hand, jumping is a physical gesture or exercise that can be related to different sports and reflects the lower body's power. In a data context, works have been developed as in [13] where a review is conducted on the meta-analysis of the effects of a plyometric jump from a female soccer player's vertical jump. Moreover, in a similar topic, the authors of [14] performed a biomechanical analysis of the effect of ball inclusion on jump performance in soccer players. These works show the interest in conducting studies to improve the performance of athletes or human evaluation. Although examples related to soccer players are shown, applications can go to Taijiquan martial art gestures [15], among others. Particularly, the vertical jump is a fundamental motor skill in many sports that influences both training and performance. For instance, when an athlete can make a short sprint or reach a high point starting from the ground, his vertical jumps make a difference [16]. Nevertheless, the vertical jump is not an easy movement. It requires coordination between the neuronal system, muscle excitation, joint motion, and force production to succeed. The main objective of the vertical jump is to bring the center of mass (COM) to the highest possible point by reaching the maximum speed in the instant of takeoff. Moreover, the jump technique could be tracked to determine its correct execution.

Previous works show an interest in generating MOCAP datasets, which allow comparative analyses or performance evaluations. Sometimes, this data type is not easy to acquire due to the cost of optical capture equipment and calibration processes. In the case of the presented work, the combination of low-cost IMUs is considered to lead to the development of low-cost technological solutions or developments that can reach a greater number of applications. It is important to highlight that the interest of this work using a low-cost IMU is to validate them in order to develop algorithms that allow their use in open field applications or under remote acquisition models. It will also allow for analyzing off-line applications and reaching users who do not have access to robust and more expensive equipment such as the Optitrack, which is used as a reference in this work.

This paper presents a dataset collected from healthy people performing a vertical jump using an IMU-based system compared to an optical MOCAP as a reference system. The use of IMUs allows the development of low-cost and customized systems for deploying applications that can be realized with similar hardware and adjusted to different areas. For this purpose, we implemented an inertial system using seven IMU devices of nine degrees of freedom, and in parallel, we used an Optitrack system configured with 22 passive markers. Both IMUs and optical markers were located at the anatomical reference points of the leg joints to obtain the anthropometric measurements and the legs' angles during the jump. This dataset is publicly available online and can be used in comparative algorithms, biomechanical studies, skeleton reconstruction, sensor fusion techniques, or machine learning models.

The rest of the paper is presented as follows. Section 2 describes the configuration of the optical system used for data acquisition, which is the reference for comparing the

inertial system. Section 3 presents the IMU-based system developed to collect the proposed dataset, giving details on the hardware and communication settings. Sections 4 and 5 show the dataset's structure and results from both acquisition systems, respectively. Finally, Section 7 gives the conclusions.

## 2. Optical System for Data Acquisition

Optical motion capture systems consist of a set of high-speed cameras that perform a three-dimensional (3D) construction of the environment based on the relative position of the cameras. The cameras detect reflective optical markers in the capture volume, and the system determines their position by triangulation, obtaining the 3D construction from image superposition algorithms.

For systems that use optical markers, the person must locate them on their body, and the system tracks the markers to reconstruct the movement. There are active and passive markers; the active ones generate light using LEDs, and the passive ones reflect the light emitted by the cameras. Systems without markers are also used, such as RGB-D cameras, which skeletonize the subject from a data array of infrared sensors. This type of system is usually less accurate. The main drawback of these systems is the markers' occlusion or falling out of the capturing volume, making it impossible to know the marker position. It is then necessary to make assumptions to perform the reconstruction. However, their accuracy is very high when optimal capture conditions are met.

In the following, we describe the configuration of the optical capture system Optitrack used to measure vertical jump performance. This system is considered the *Gold Standard* in the field of motion capture.

### 2.1. Hardware

The optical capture system manufactured by Optitrack$^{TM}$ was used as the reference system. The system includes six Flex 3 cameras at 100 FPS, linked to two synchronized OptiHub with a USB connection to a computer. Figure 1 shows the used devices. These devices provide a compact approach with all required features for image capturing and motion tracking. The optical system has an accuracy of $0.2 \pm 0.01$ mm.

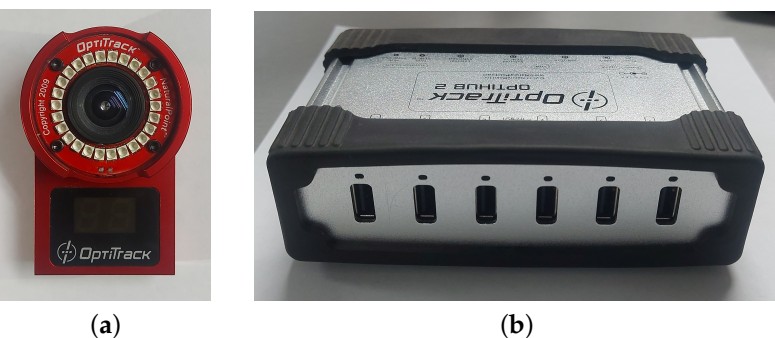

(**a**) (**b**)

**Figure 1.** Devices used in the optical capture system: (**a**) Optitrack Flex 3 camera; (**b**) Optitrack OptiHub devices, where the cameras are connected to the computer.

Motion capture was performed using Optitrack's Motive Tracker software. Figure 2 depicts the capturing area setup, whose the capture volume has 2 m diameter and 2 m height (sufficient volume for capturing the human lower body). The cameras are installed in two parallel rails, three cameras and a hub per rail. Both hubs are synced with an RCA cable. This installation generates a semi-circular capture volume. The ground level capture area is shown in Figure 2b drawn with black tape. The highest possible point for capture is set by the camera height.

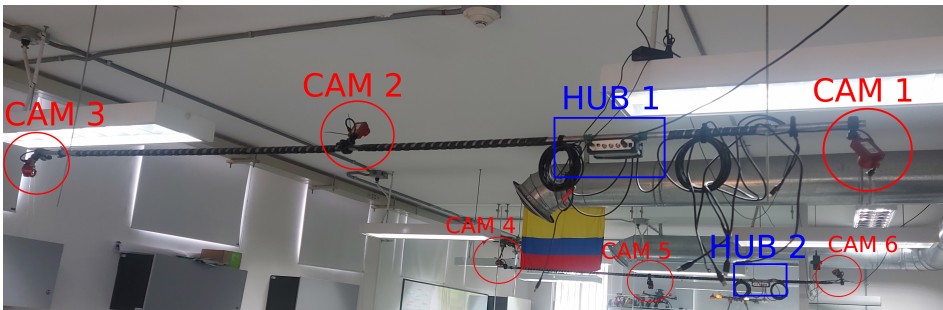

(**a**) Camera setup. In red circles are the Flex 3 cameras. In blue boxes are the OptiHub devices.

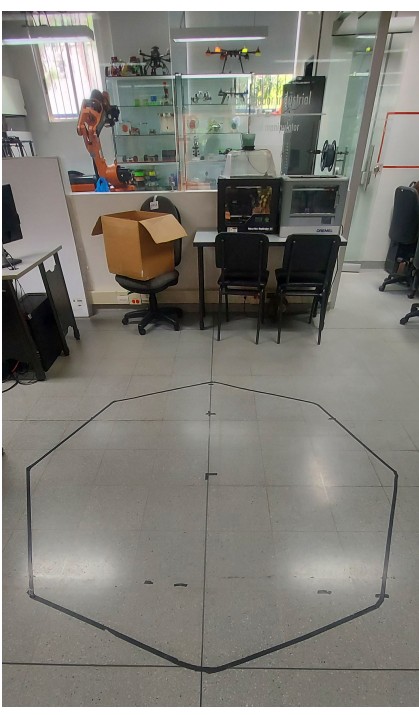

(**b**) Capture area.

**Figure 2.** Representation of the capture volume of the optical capture system.

### 2.2. Calibration

The calibration process consists of two stages. In the first stage, the relative position between the cameras is calculated, and in the second one, the ground reference plane is determined.

For the first stage, we used the CW-250 calibration tool (Figure 3a), which has three markers aligned at different and known distances. The CW-250 is placed in the capturing area and moved to try to cover the largest possible volume. Thus, for each image obtained from the markers, a spatial relationship is established with the distance between them. After having a sufficient number of samples, a three-dimensional reconstruction can be obtained; therefore, the greater the number of samples, the greater the precision is.

The calibration process continues using the CS-200 tool (Figure 3b) in the second stage. The CS-200 has three markers in an orthogonal arrangement forming a plane. This arrangement becomes the capture volume origin and determines the floor plane. It is located in the volume's center, and all cameras take a simultaneous image. Thereby, with the spatial distribution obtained from the first stage, the CS-200 position is established to determine the origin of the capture volume. After executing these two procedures, the markers can be captured, and their position respecting the volume origin can be established. Typical estimation errors are within $0.2 \pm 0.01$ mm, which determine the system's accuracy.

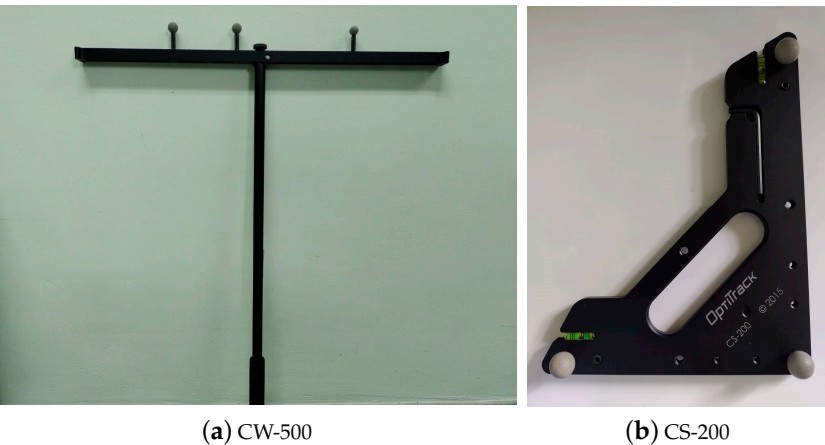

(**a**) CW-500          (**b**) CS-200

**Figure 3.** Calibration tools used for the optical capture system.

*2.3. Marker Locations*

Since the objective is to obtain and compare the anthropometric measurements and the legs' angles with the inertial system, the markers are located at the anatomical reference points of the leg joints. In the rigid segments, they are located asymmetrically with respect to the two legs, which makes it possible to differentiate them. Thus, in addition to the position of the markers, the joints' angles can be obtained. Figure 4 shows the placement of the reflective optical markers on the legs, represented as yellow squares, alongside the IMU positions as cyan circles.

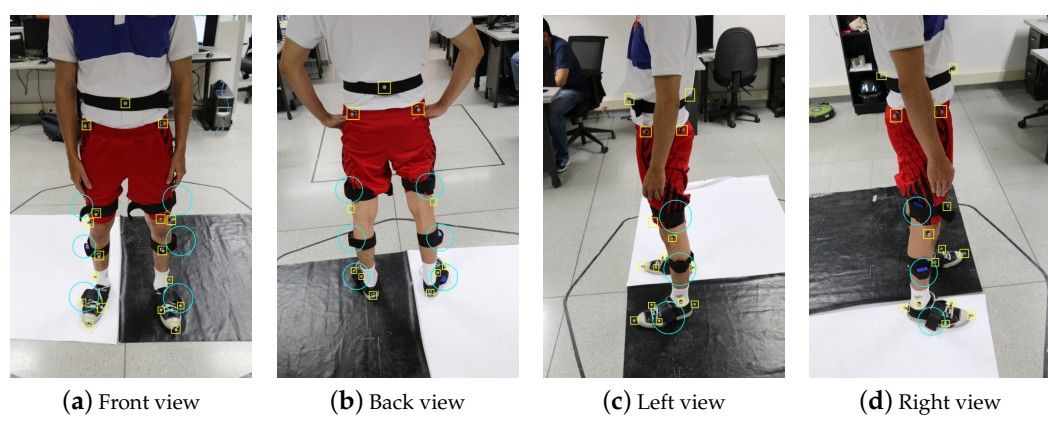

(**a**) Front view     (**b**) Back view     (**c**) Left view     (**d**) Right view

**Figure 4.** Configuration of the optical markers' and IMUs' placement for measuring a vertical jump. The reflective markers for the optical capture system are shown in the yellow squares. IMUs are placed in cyan circles for the inertial system.

## 3. Proposed IMU-Based Inertial System

Accelerometers and gyroscopes are commonly known as inertial sensors. An Inertial Measurement Unit (IMU) is basically composed of the combination of three orthogonal accelerometers and three orthogonal gyroscopes. On the one hand, the orthogonally aligned accelerometers allow obtaining the absolute acceleration vector of the body to which it is attached. With this data, it is possible to perform integration with respect to time and obtain the velocity vector and then a second integration to obtain the body's position in relation to a reference. Secondly, the absolute orientation of the body can be obtained with the gyroscope arrangement that measures the rotation rate (degrees per time unit) in each axis, also synchronized to the time. Thus, IMUs can provide information on the position and orientation of a body or object. These sensors can be attached by other sensors such

as magnetometers or GPS to obtain redundant information that improves the accuracy of the systems.

In recent years, the development of Micro-Electromechanical Systems (MEMS) has reduced the inertial sensor dimensions. Thus, IMUs can be used in several applications, such as human motion capture. In the following, we describe the system developed based on IMUs to obtain the position and orientation of the body's lower segments during vertical jumps. It is worth mentioning that the system is portable and does not affect the exercise execution.

### 3.1. Hardware

The inertial acquisition system was designed to be compact and lightweight. Thus, it is possible to locate it on the person's body without generating inconvenience while performing movements. Figure 5 depicts the general scheme of the proposed IMU-based inertial system and samples of system components. The dimensions of the assembled device are 4.0 cm high, 2.5 cm wide, and 4.4 cm in long. The proposed system has five main parts. The features are described below. features :

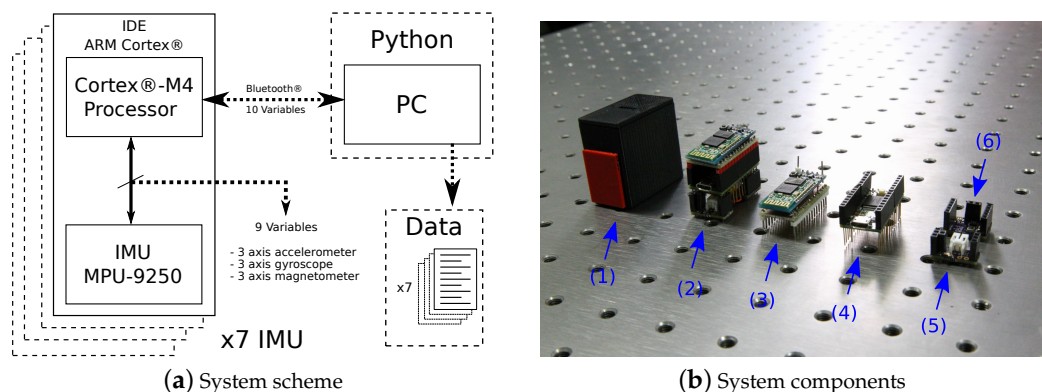

(**a**) System scheme  (**b**) System components

**Figure 5.** Proposed IMU-based inertial system: (**a**) general scheme depicting components and communication links; and (**b**) samples of device and components, from left to right: (1) assembled device; (2) device without enclosure; (3) Bluetooth module; (4) microcontroller; (5) charging circuit; and (6) IMU MPU-9250.

1. Microcontroller: a 32-bit 72 MHz ARM Cortex-M4 32-bit MK20DX256VLH7 from Freescale Semiconductor, mounted on a Teensy 3.2 development board.
2. IMU MPU-9250: combines a triaxial accelerometer, a triaxial gyroscope, and a triaxial magnetometer for motion monitoring, enclosed in a $3 \times 3 \times 3$ mm chip developed by InvenSense [17]. These sensors have programmable operating ranges as follows: the accelerometer has operating scales of $\pm 2$, $\pm 4$, $\pm 8$, and $\pm 16$ gravity (*g*), the gyroscope $\pm 250$, $\pm 500$, $\pm 1000$, and $\pm 2000$ degrees/second, and the magnetometer $\pm 1200$ micro-tesla.
3. Bluetooth module: the 2.4 GHz Bluetooth HC-05 serial module is used. The transfer rate was set to $57,600$ Bd.
4. Battery charging circuit: to ensure that each IMU does not require connection to an external power source, it is powered by a 110 mAh 3.7 V lithium-ion battery. Additionally, a circuit is responsible for charging the battery by supplying 5 V from a USB connection and powering the other modules.
5. Personal computer: DELL T5610, Inter Xeon E5-2667 24 cores 2.90 GHz, 32 Gb RAM DDR3.

### 3.2. Communication

Each IMU has a Bluetooth module connected to a receiver on a personal computer for communication and data transmission. Data are acquired simultaneously from all seven

IMUs and stored in text files. For each sample, 10 data variables are sent: acceleration in *X*, *Y*, and *Z* axes; rotation speed in *X*, *Y*, and *Z* axes; magnetic field in *X*, *Y*, and *Z* axes; and the microcontroller timestamp corresponding to the sampling time. Notice that the timestamp is not the collection time but a reference to the capturing moment. The sensor data are a 32-bit floating number. To reduce the number of characters sent, each value is converted to hexadecimal using the IEEE Standard for Floating-Point Arithmetic (IEEE 754) [18]. Thus, we have 72 characters for the sensors and 4 for the sampling time, which is a 16-bit integer. Table 1 shows a description of a data frame sample sent from an IMU.

**Table 1.** Description of a data frame sample sent by serial Bluetooth communication from an IMU. The last row shows the complete data frame.

| Data Variable | | Hexadecimal Value | Decimal Value | Unit |
|---|---|---|---|---|
| Accelerometer | *X*-axis | 3F1CEA16 | 0.61294686794281005859375 | |
| | *Y*-axis | BFACD9DC | −1.350398540496826171875 | m/s$^2$ |
| | *Z*-axis | C11670AD | −9.40250873565673828125 | |
| Gyroscope | *X*-axis | 3C3BA031 | 0.01145176682612406158447265625 | |
| | *Y*-axis | 3C8FFDEA | 0.0175771303474903106689453125 | deg/s |
| | *Z*-axis | 3B945AF1 | 0.00452744262292981147766113281250 | |
| Magnetometer | *X*-axis | C20A9F8D | −34.655811309814453125 | |
| | *Y*-axis | 4259EFAE | 54.48406219482421875 | mT |
| | *Z*-axis | 4254CBD2 | 53.19904327392578125 | |
| Timestamp | | 51FC | 20,988 | ms |
| Dataframe | | 3F1CEA16BFACD9DCC11670AD3C3BA0313C8FFDEA3B945AF1C20A9F8D4259EFAE4254CBD251FC | | |

Since the communication is serial Bluetooth, the size of each sample is 8 × 72 bits (576 bits), having a floating-point precision of 16 bits. Each IMU sends data to the receiver at a rate of 57,600 baud. The sensors are sampled by an Inter-Integrated Circuit bus (*I*$^2$*C* for Inter-Integrated Circuit) at a frequency of 1 kHz. After transmission, the data are read at 75 Hz, a rate close to the traditional sampling rate for human motion [19–21]. After receiving the data, they are decoded and stored in text files as explained in Section 4.

### 3.3. IMUs Location

To capture motion from inertial sensors, each limb of the body (in this case, the lower body) is treated as an independent rigid element; therefore, an IMU is attached to each segment. Figures 4 and 6 show the IMU locations arranged to measure a vertical jump. Aiming to avoid discomfort when performing the exercise, we located the devices outside each leg and at the back of the waist. The location is arbitrary since the proposed method does not depend on a specific location since it is estimated from the initial data of the accelerometers and magnetometers.

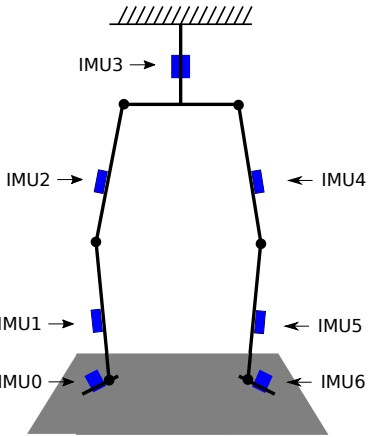

**Figure 6.** Scheme of kinematic chain model of the lower body, the blue boxes represent the IMUs locations.

## 4. Data Description and Structure

The dataset comprises eight jumps recorded from four healthy humans using the optical reference system and the proposed IMU-based system. The four participants where male, with good physical condition, age between 20 and 30 years. All participants performed five jumps with one minute rest between them. After the jumps were recorded, we cleansed the data. The proposed database is publicly available online at the Zenodo repository through the following link: 10.5281/zenodo.6600752. The structure of the dataset is as follows:

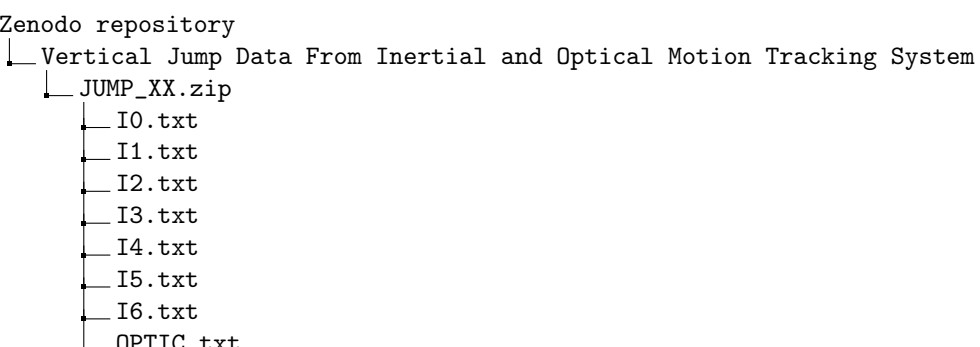

```
Zenodo repository
└── Vertical Jump Data From Inertial and Optical Motion Tracking System
    └── JUMP_XX.zip
        ├── I0.txt
        ├── I1.txt
        ├── I2.txt
        ├── I3.txt
        ├── I4.txt
        ├── I5.txt
        ├── I6.txt
        └── OPTIC.txt
```

Each jump is stored as a separate `JUMP_XX.zip` file, where XX corresponds to the jump number. `OPTIC.txt` contains the data from the optical motion tracking system; it includes the XYZ position of each optical marker. TXT files store data variables from the IMUs, and the order is shown in Figure 6: `I0.txt` from Right Foot, `I1.txt` from Right Lower Leg, `I2.txt` from Right Upper Leg, `I3.txt` from Center Of Mass (waist), `I4.txt` from Left Upper Leg, `I5.txt` from Left Lower Leg, and `I6.txt` from Left Foot.

## 5. Data Samples

This section shows samples for a jump captured with the optical and inertial systems contained in the dataset. For conciseness, only the data for a single jump are shown. For the optical system data, skeletonization is made from the location of the markers. The inertial system data are shown in their raw format as supplied by the sensors, without any processing.

### 5.1. Optical System

After running the calibration procedure of the cameras in the Optitrack Motive Tracker Software, the program has the spatial location of each camera. With their locations, the system can reconstruct a volume in the capture space. Thus, the position of the markers is obtained. After recording a jump, the software exports each marker position captured in the volume. We used the markers located at the joints or biometric points to construct the skeletonization of the subject, connecting them and forming the links. The angles at the joints are calculated using geometry from the segments obtained. Figure 7 shows the skeletonization obtained from the data provided by the Optitrack system at different jump moments.

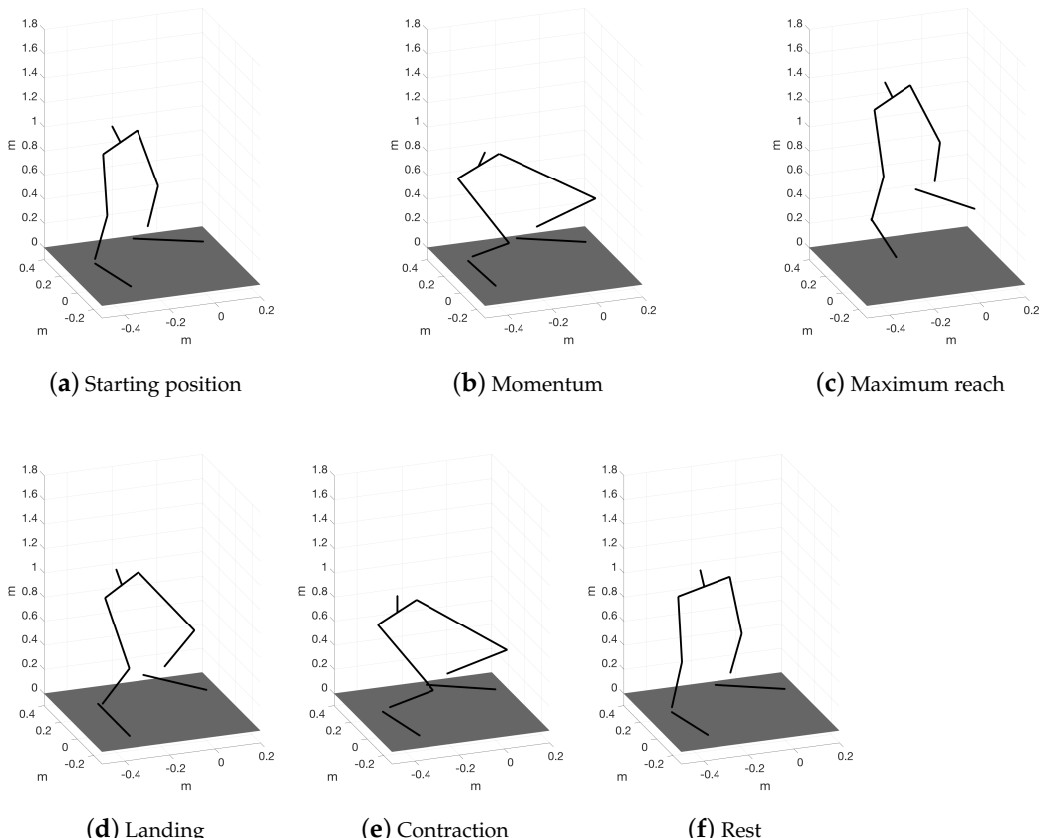

(**a**) Starting position

(**b**) Momentum

(**c**) Maximum reach

(**d**) Landing

(**e**) Contraction

(**f**) Rest

**Figure 7.** Skeletonization samples for a jump captured by the optical motion capture system from markers in anatomical positions.

*5.2. Inertial System*

As explained in the previous section, there are seven text files for each recorded jump, one for each IMU. Each file contains the sensors' hexadecimal data, containing the signals provided by accelerometers, gyroscopes, and magnetometers. Plots for the raw data obtained by the inertial system are shown below. Figures 8–10 show the data provided by the accelerometers, gyroscopes, and magnetometers, respectively, from each IMU during the entire execution of an example jump. It is worth mentioning that the signals shown are from the captured raw data (with the digital motion processing (DMP) features enabled), and no pre-processing or noise reduction was applied.

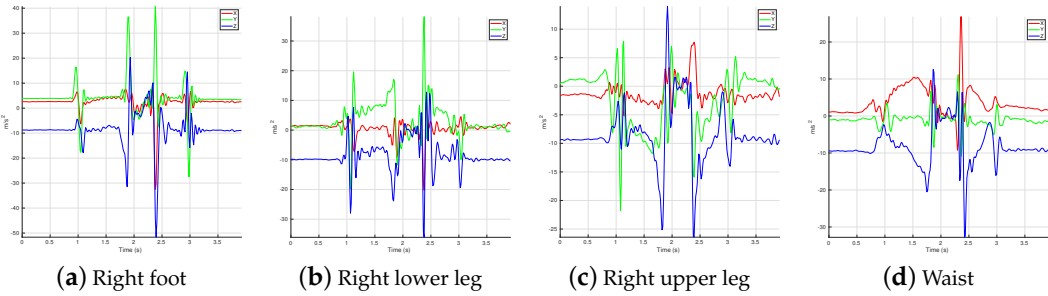

(**a**) Right foot

(**b**) Right lower leg

(**c**) Right upper leg

(**d**) Waist

**Figure 8.** *Cont.*

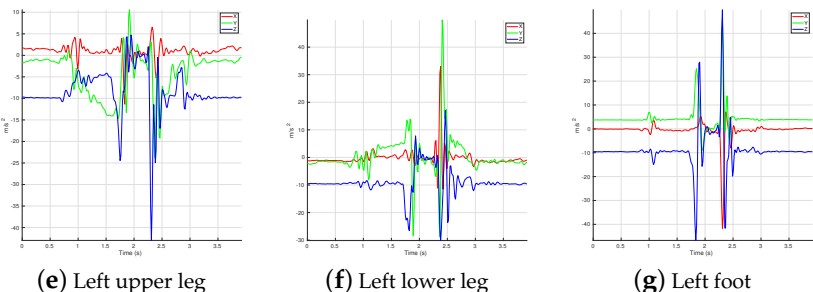

**Figure 8.** Accelerometer signal samples from each IMU for a jump captured by the inertial system.

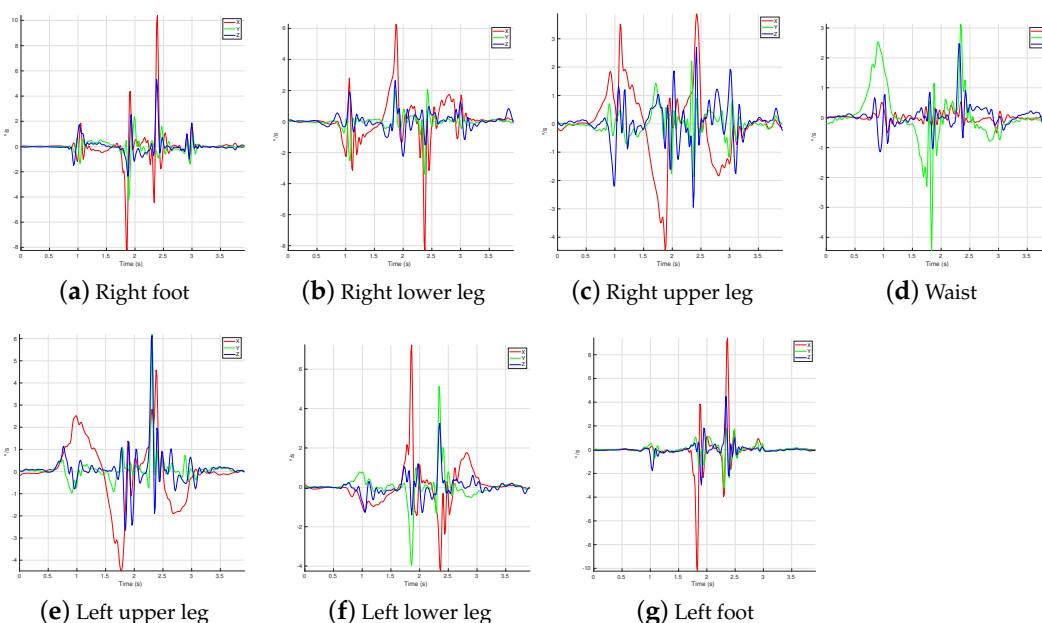

**Figure 9.** Gyroscope signal samples from each IMU for a jump captured by the inertial system.

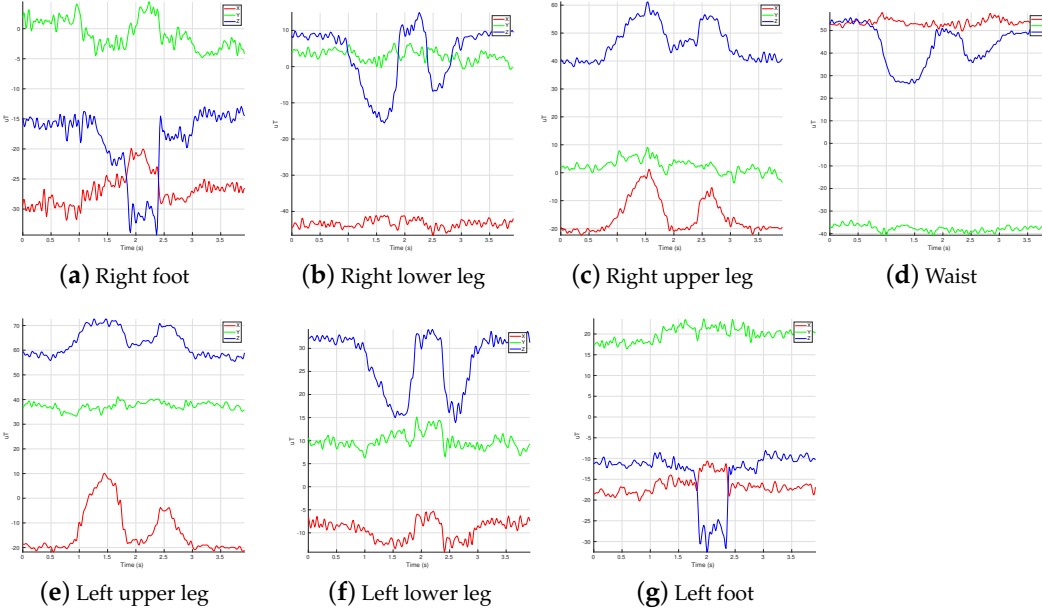

**Figure 10.** Magnetometer signal samples from each IMU for a jump captured by the inertial system.

## 6. Dataset Usage Examples and Results

The proposed examples for jump measuring using the dataset are composed of two stages. Firstly, it consists of calculating each IMU orientation using a quaternion-based operation; with this information, the joint angles are determined. The second stage is for determining the COM's speed and position. The initial orientation of each IMU is calculated from the pseudo-orthogonal system between the gravity vector and the magnetic field.

### 6.1. Joint Angles

Each body segment to which an IMU is attached has an initial unit vector ($\vec{I}$) for orientation, given by the pseudo-orthogonal system between the gravity vector and the magnetic field. From this director vector, the orientation changes are made using the gyroscope data. A unit quaternion from the angular velocity is calculated using Equation (1).

$$q = \left[ \cos\left( \|\omega\| \Delta t \right) \quad \frac{\omega}{\|\omega\|} \sin\left( \|\omega\| \Delta t \right) \right] \tag{1}$$

where $\omega$ is the three-dimensional vector of gyroscope data, and $\|\omega\| \Delta t$ is the angle $\theta$ of rotation. Using $q$, the initial unit vector ($\vec{I}$) is rotated with Equation (2):

$$\vec{I}_r = q\vec{I}\bar{q} \tag{2}$$

Thus, the joint angles are defined as the angles between the director vectors. Figure 11 illustrates the rotation process of each body segment.

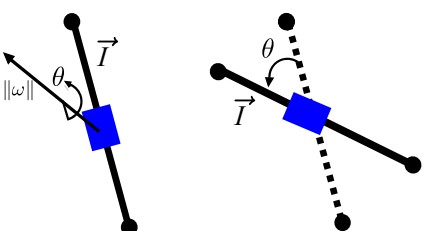

**Figure 11.** Illustration of the body segments rotation using the gyroscope data and the quaternion approach.

Following, we show data processing results to obtain the bio-mechanical values on a vertical jump from the dataset. The ankle, knee, and hip joints are calculated with the unit vector of each segment. Figure 12 shows the angles of each joint in a vertical jump. These values were compared with the optical system and are shown in Table 2.

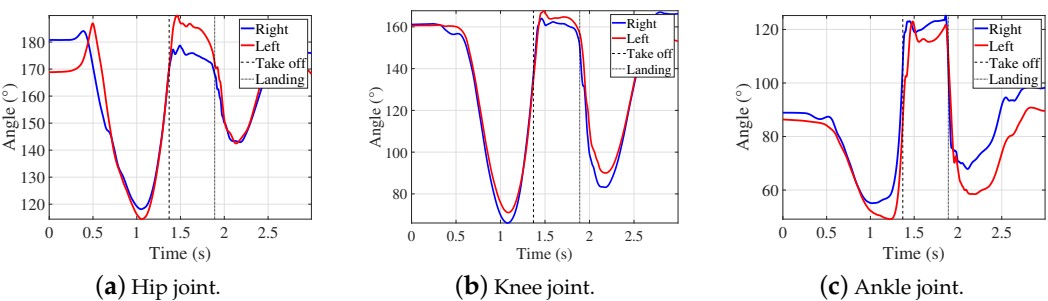

(**a**) Hip joint.   (**b**) Knee joint.   (**c**) Ankle joint.

**Figure 12.** Joint angles in the vertical jump measured with the IMUs. The takeoff and the landing time is marked with vertical lines.

**Table 2.** Comparison between both estimation systems of the bio-mechanical variables involved in the vertical jump. The value *r* indicates the correlation between the signals of each system. The mean and standard deviation (SD) are shown for each system.

| Measurement | | r | Optitrack | | IMU | | Units |
|---|---|---|---|---|---|---|---|
| | | | Mean | SD | Mean | SD | |
| Ankles | Right | 0.989 | 79.222 | 21.113 | 83.094 | 13.751 | ° |
| | Left | 0.973 | 77.648 | 22.616 | 95.863 | 12.550 | ° |
| Knees | Right | 0.992 | 141.020 | 25.021 | 136.421 | 23.757 | ° |
| | Left | 0.999 | 134.999 | 29.490 | 136.852 | 24.999 | ° |
| Hips | Right | 0.812 | 32.979 | 34.868 | 21.670 | 12.818 | ° |
| | Left | 0.89 | 30.011 | 28.837 | 32.707 | 13.711 | ° |
| COM | Axis *X* | 0.30 | −0.044 | 0.043 | 0.007 | 0.029 | m |
| | Axis *Y* | 0.70 | 0.058 | 0.132 | −0.001 | 0.155 | m |
| | Axis *Z* | 0.79 | −0.001 | 0.005 | 0.005 | 0.022 | m |

*6.2. COM Position*

An IMU is placed near the COM (IMU3 in Figure 6), at the waist near the fifth lumbar vertebra (L5 vertebra) [22]. To calculate the COM's speed and position, the gravity acceleration is removed from the accelerometer measurements depending on the IMU rotation, as described in the previous section. Figure 13 shows the jump sequence and the rotation of the IMU. Numerical integration to the 3D acceleration gives the 3D velocity. This operation adds up bias because the area under the curve in the movement is not zero or close. Nonetheless, the speed at the beginning and end of the movement is zero; with this assertion, the bias of the operation can be determined. A threshold is set in the acceleration magnitude signal to get its duration. A Single Rectangular Pulse of Movement (SRPM) is obtained, where active SRPM indicates the movement duration. In the active SRPM, numerical integration is applied to the 3D acceleration using the trapezoidal rule. In the inactive SRPM, the integral result is set to zero. This operation is shown in Equation (3).

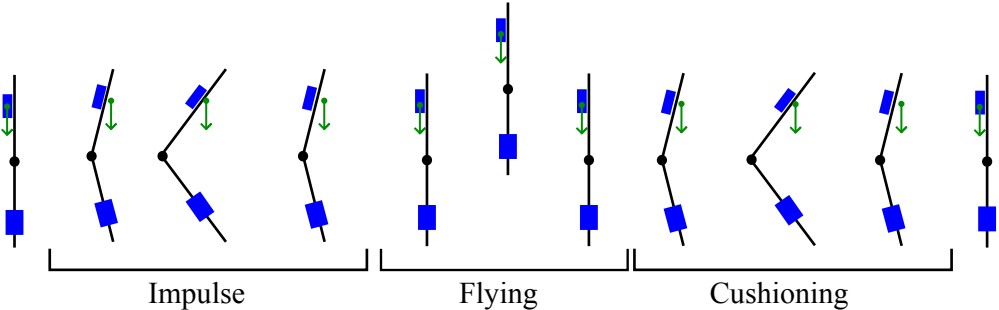

**Figure 13.** Jump sequence of the trunk and the thigh. The rotation of the waist IMU is shown with reference to the gravity vector (green arrow).

$$
\vec{v}_t = \begin{cases} \vec{v}_{t-1} + \left( \dfrac{\vec{a}_t + \vec{a}_{t-1}}{2} \right) \Delta t & \text{if} \quad \|\vec{a}_t\| > \text{threshold} \\ 0 & \text{if} \quad \|\vec{a}_t\| \le \text{threshold} \end{cases} \tag{3}
$$

where $\vec{v}_t$ is the velocity result, and $\vec{a}_t$ the acceleration vector. At the trailing edge of the SRPM, the velocity result must be zero, so the bias will be the integral value in that instant of time. The velocity bias of each sample is calculated using Equation (4).

$$
\mathbf{S}_k = k \left( \frac{\vec{v}(n)}{n} \right) \quad k = 1, 2, 3, 4, \ldots, n \tag{4}
$$

where *n* is the number of samples in the duration of movement, and *k* the actual sample. Finally, $\mathbf{S}_k$ is subtracted in $\vec{v}_t$.

For position calculation, the above assertion cannot be made, i.e., the final position is not zero because the jumper does not return to the beginning spot. For this reason, just the trapezoidal rule is applied.

In the following, we show a performance analysis with the obtained data from joint angles and COM position. Figure 14a depicts the typical signal of an accelerometer located at the waist during a vertical jump. From these signals, the gravity is removed depending on the IMU rotation. Figure 14b represents the signals after gravity subtraction. The signal $\|\vec{a}_t\|$ and the binary signal of the movement interval are shown in Figure 15. Afterwards, Equation (3) is applied, and the results are shown in Figure 16a. The bias subtraction from the 3D velocity result is shown in Figure 16b.

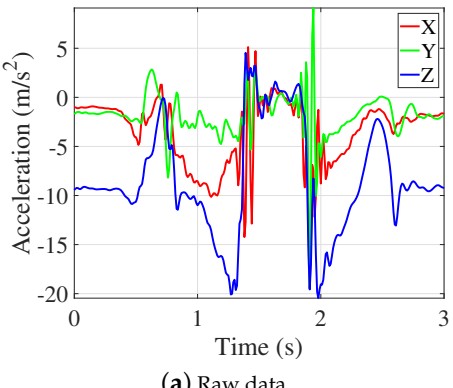
(**a**) Raw data.

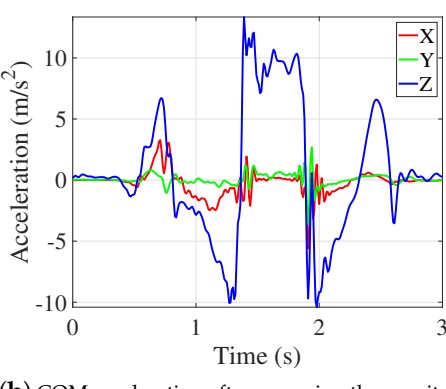
(**b**) COM acceleration after removing the gravity

**Figure 14.** Three-dimensional acceleration of the COM in a squat vertical jump measured with the IMU at the waist.

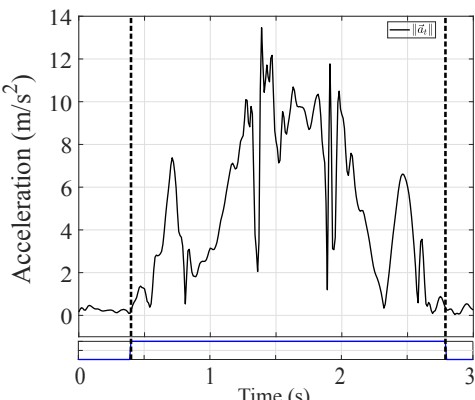

**Figure 15.** Magnitude of the three-dimensional acceleration of the COM and the binary signal (blue line) of the movement interval after the threshold operator application. Dashed lines indicate the start and end of the jump.

In the velocity result, the flying time can be easily estimated. The maximum ascending speed denotes the instant of takeoff; from there, the velocity starts to descend linearly to the moment of landing. The maximum height is calculated with this data and the free-fall equations. In this study, the validity of the inertial system in COM tracking in the vertical jump must be tested against the optical system. For this, integration is made in the COM velocity signal to obtain the COM position. The results and the comparison of both systems are shown in Figure 17, and the Table 3.

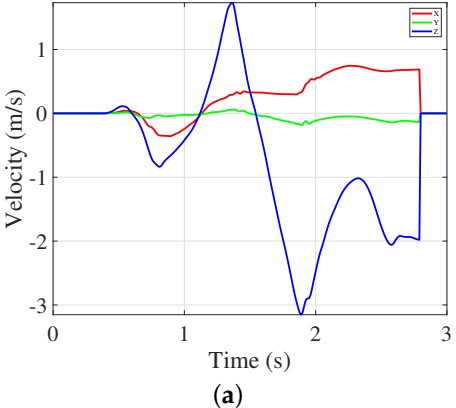
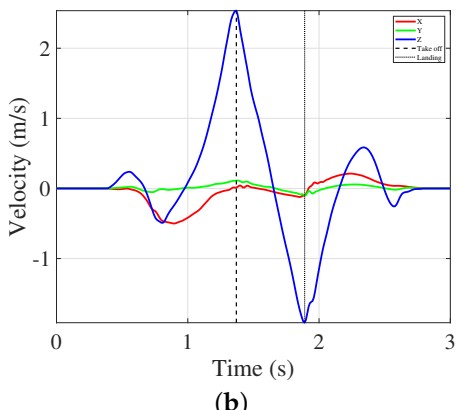

**Figure 16.** Three-dimensional velocity obtained from the numerical integration using the movement intervals: (**a**) three-dimensional velocity before removing the bias; (**b**) three-dimensional velocity after removing the bias.

**Table 3.** Maximum height estimation. Comparison between both systems.

| Measurement | Optitrack | IMU | % Error |
|---|---|---|---|
| Max height | 0.3837 m | 0.3921 m | 2.1908 |

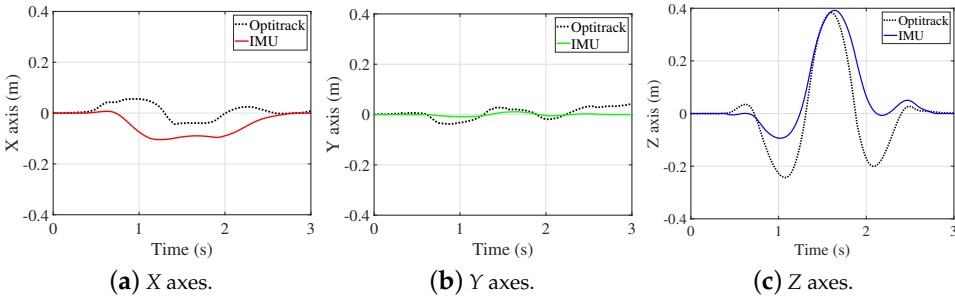

**Figure 17.** Three-dimensional position of COM.

With the proposed application to measure the maximum height reached in the jump, high accuracy was obtained. It is an important measure for a future limb power estimation system. Likewise, the joint angles in Figure 12 provide information about which muscular power is generated and dissipated using the lower train. Moreover, the developed approach can be applied in research about bipedal locomotion such as gait analysis, sit–stand transfers, and step-up transfers [23,24].

## 7. Conclusions

This paper presented a dataset including optical motion capture system data and inertial measurement capture system data from eight vertical jumps. The proposed dataset was designed to fundamentally compare both systems, validating algorithms that use the inertial data and comparing with the optical data from the Optitrack system, which is considered the gold standard in motion capture systems. The dataset is publicly available online, and the data are very simple to use since the file names link both system data.

The main contribution of the proposed IMU-based approach is that it allows the user to try and design algorithms related to human motion tracking without buying and setting up expensive and delicate systems. Moreover, since the proposed dataset was collected using an IMU-based system and an optical MOCAP as a reference system, it contains correlated data that allow us to compare an IMU-based low-cost system against the MOCAP gold standard. In contrast, the main disadvantage of the proposed method is that it requires a controlled environment and the collaboration of the subject.

As future work, we plan to extend the size of the dataset, incorporating the study of a larger number of subjects. In addition, statistical characterization of the dataset could be of interest depending on the study application. Further studies can be carried out on biomechanical studies, skeleton reconstruction, sensor fusion techniques, or machine learning models. New versions of the database will be available in the same public repository.

**Author Contributions:** Conceptualization, M.R.-G., J.B.-V. and R.H.-G.; methodology, M.R.-G. and J.B.-V.; validation, R.H.-G.; writing—original draft preparation, M.R.-G., J.B.-V. and R.H.-G.; writing—review and editing, M.R.-G. and R.H.-G.; funding acquisition, J.B.-V. and R.H.-G. All authors have read and agreed to the published version of the manuscript.

**Funding:** This research received no external funding. The APC was funded by Universidad Católica del Maule.

**Institutional Review Board Statement:** The study was conducted in accordance with the Declaration of Helsinki. Ethical review and approval were not required, but informed consent was obtained from all subjects involved in the study.

**Informed Consent Statement:** Informed consent was obtained from all subjects involved in the study.

**Data Availability Statement:** Mateo Rico-Garcia, Juan Sebastian Botero-Valencia, & Ruber Hernández-García. (2022). Vertical Jump Data From Inertial and Optical Motion Tracking System. DOI: 10.5281/zenodo.6600752 (accessed on 10 August 2022).

**Acknowledgments:** This study were supported by the Sistemas de Control y Robótica (GSCR) Group COL0123701, at the Sistemas de Control y Robótica Laboratory, attached to the Instituto Tecnológico Metropolitano. R.H.-G. also thanks to the Research Project ANID FONDECYT Iniciación en Investigación 2022 No. 11220693.

**Conflicts of Interest:** The authors declare no conflict of interest.

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
