# Peer review of "Vertical Jump Data from Inertial and Optical Motion Tracking Systems"

_data, 2022_

Round 1

Reviewer 1 Report

Although the manuscript contains important results, the number of references is relatively few and the study lacks comparison with previous works.

Author Response

We would like to thank you for your kind and valuable comments. Please, find attached the responses.

Reviewer 2 Report

The authors aim to share a small dataset that combines Optical Motion and Inertial Systems, capturing a well-known sports movement as the vertical jump.  The article is properly written and has a smooth logical flow. However, some parts of the methodology are unclear which weakens the authors’ conclusions.  I have some remarks for the authors to improve their manuscript. I have a list of some of my questions below:

1. The manuscript is focused on the vertical jump, but the reason is not clear, for example, what are the advantages and disadvantages of the vertical jump, and why is this small database enough to contribute to the scientific community? 

see for example https://journals.lww.com/nsca-jscr/Fulltext/2017/01000/Traditional_vs__Sport_Specific_Vertical_Jump.22.aspx 

2. The dataset can be used in comparative algorithms, biomechanical studies, skeleton reconstruction, sensor fusion techniques, or machine learning models. It would be interesting to expand examples of these applications, It is possible to develop someone in the paper as an example? Another important issue is about machine learning, how do you justify that studies of this type can be carried out with a small dataset? see https://www.computer.org/csdl/proceedings-article/icpr/2021/09412492/1tmjFUbughy .

3. The IMU-based system must be emphasized in the abstract and the introduction since it is the main contribution after the dataset.

4. With respect to Figures 8 to 10 

a. Are the time signals free of noise? Is the acquisition of the signal really free of noise? was any filter stage used?

b. What are the variances of these figures between them, and with respect to the resting stage? 

5. With all the information obtained in section 5, is it possible to characterize the dataset?

6. A section of results is necessary!

7. It is very important to determine the general advantages and disadvantages of the manuscript to improve the main contribution of this work.

8. Minor points:

a. Latex has a package to cut words correctly, e.g. for the "comparative" word in line 50, you can use \hyphenation{com-pa-ra-ti-ve}.

b. The year of reference 5 is wrong, is it 2013?

Author Response

(The authors gave the same response as above.)

Reviewer 3 Report

This paper introducing a vertical jumps dataset that combines optical motion and inertial systems.

In line 66-68, it is mentioned that the dataset is publicly available online and can be used in other models, the authors can explain, if it is possible to use it in fall detection for example. In the case of machine learning models, how are they intended to be used? Set a section for future work in the document.

Figure 2. Representation of the capture volume of the optical capture system, is very small, it is not possible to distinguish, it is recommended to make it a little larger. In addition, the camera number must be added in the image to distinguish the camera and know the number; although there are few cameras, it is better for the reader.

Figure 3. Calibration tools used for the optical capture system, it is not visible in the document, is it a problem with the pdf?

Figure 5 (b), the authors are asked to place a label indicating each of the parts, it is true that in the description of the figure it is shown, but for the reader it is reasonable to have it in the figure.

In table 1, the timestamp is 20988 ms? this is a long time, isn't it? it is necessary to explain this information well, to avoid misinformation. Is it possible to reduce the data collection time? How can this be done?

We are grateful to the authors for making the dataset available online and for the explanation of its structure.

Author Response

(The authors gave the same response as above.)

Round 2

Reviewer 3 Report

The authors have correctly completed what has been requested.